# Single molecule compression reveals intra-protein forces drive cytotoxin pore formation

**Daniel M Czajkowsky, Jielin Sun, Yi Shen, Zhifeng Shao\***

State Key Laboratory of Oncogenes and Related Genes and Bio-ID Center, School of Biomedical Engineering, Shanghai Jiao Tong University, Shanghai, China

**Abstract** Perfringolysin O (PFO) is a prototypical member of a large family of pore-forming proteins that undergo a significant reduction in height during the transition from the membrane-assembled prepore to the membrane-inserted pore. Here, we show that targeted application of compressive forces can catalyze this conformational change in individual PFO complexes trapped at the prepore stage, recapitulating this critical step of the spontaneous process. The free energy landscape determined from these measurements is in good agreement with that obtained from molecular dynamics simulations showing that an equivalent internal force is generated by the interaction of the exposed hydrophobic residues with the membrane. This hydrophobic force is transmitted across the entire structure to produce a compressive stress across a distant, otherwise stable domain, catalyzing its transition from an extended to compact conformation. Single molecule compression is likely to become an important tool to investigate conformational transitions in membrane proteins.

## Introduction

The initial demonstration that single macromolecules could be mechanically stretched with exact knowledge of the associated forces (*Smith et al., 1992*; *Kellermayer et al., 1997*; *Rief et al., 1997*; *Tskhovrebova et al., 1997*) has enabled unprecedented access to measurements of the physical and chemical inter-molecular interactions (*Bippes and Muller, 2011*; *Bustamante et al., 2000*; *Fisher et al., 2000*; *Hu and Li, 2014*; *Zhuang and Rief, 2003*). Moreover, pulling on single molecules with tensile forces has enabled detailed probing of functional and structural transitions within proteins, in particular those that involve a separation of specific structural elements (*Puchner and Gaub, 2012*; *Zhang et al., 2009*; *del Rio et al., 2009*). Yet, tensile forces cannot be used to probe transitions that involve a decrease in the distance between structural elements such as the closing of a substrate access gate, for example, as the force acts in a direction opposite to the structural movement. In these instances, what is needed is the controlled application of a compressive force at the single molecule level.

Perfringolysin O (PFO) is an intriguing pore-forming toxin that undergoes a significant reduction in the distance between two of its domains during pore formation (*Gilbert, 2005*; *Hotze and Tweten, 2012*). During the prepore-to-pore transition, the distance between the D1 and D4 domains is reduced as a result of conformational changes in the D2 domain (*Figure 1*): this domain is elongated in the prepore but collapses into a more compact structure in the pore, largely accounting for the remarkable 40 Å decrease in height relative to the bilayer surface first observed by atomic force microscopy (AFM) (*Czajkowsky et al., 2004*; *Tilley et al., 2005*). Also within this prepore-to-pore step, an α-helical bundle (the Trans-Membrane Helices (TMHs)) in the D3 domain converts into an amphipathic β-sheet that inserts in the bilayer and lines the aqueous membrane pore.

**\*For correspondence:** zfshao@sjtu.edu.cn

**Competing interests:** The authors declare that no competing interests exist.

**eLife digest** Proteins are made up of chains of amino acids that need to fold into intricate three-dimensional shapes to work correctly. But some proteins also have to change their shape drastically when they work. Mechanical forces that change the shape of a protein can therefore be used to determine how a protein folds and how it changes its structure when working.

Although researchers have developed techniques to analyze the effect of force on single proteins, most studies carried out so far have investigated the effect of stretching (or tensile forces) to understand structural changes that naturally involve an extension within the protein. However, many proteins undergo structural changes that involve a compaction in their shape. How these changes occur remains poorly understood because, for these, methods to apply compressive forces to single proteins are required.

Perfringolysin O (PFO for short) is a protein that is made by a bacterium that causes food poisoning in humans. PFO makes pores in the membrane that surrounds cells. This causes the cell's contents to leak out, killing the cell. When inserting into the membrane, PFO changes from an elongated "prepore" state to a compact pore-forming state.

Czajkowsky et al. now use a combination of single molecule techniques and computer simulations to investigate how PFO undergoes this compaction. Previous work had identified a mutant PFO protein that arrests at the prepore state. Applying a compressive force to the top of this prepore-trapped PFO as it sits on the membrane transmitted forces across the entire PFO protein. This ultimately produced a compressive force across a distant part of the protein that caused the protein to change from the elongated prepore state to the compact, pore-like shape. If a compressive force was not applied, the PFO protein remained in the prepore state. Czajkowsky et al. further found that this compressive force is naturally produced by distant water-repellent parts of the naturally occurring protein interacting with the cell membrane. Therefore, internal forces can transmit across proteins to drive shape changes in distant regions.

In the future, the methods developed in this study could be applied to analyze other naturally occurring changes in proteins where shape compaction happens when working.

While these changes in the D2 and D3 domains have been firmly established, the mechanism by which these changes are coordinated remains unresolved. In fact, the physical processes associated with long-distance structural communication in proteins are poorly understood in general (*Cui and Karplus, 2008*; *Whitley and Lee, 2009*; *Li et al., 2011*). Herein, using single molecule compressive force spectroscopy and single molecule AFM together with all-atom molecular dynamics (MD) simulations, we demonstrate that intra-protein stresses are the driving force of the structural coordination between the D2 and D3 domains and, ultimately, the catalyst of the collapse of the D2 domain.

## Results and discussion

As the D2 and TMHs directly contact each other in the PFO prepore and are well separated in the pore, it was previously suggested that the loss of these contacts during the prepore-to-pore transition might have precipitated the collapse of an inherently unstable D2 domain (*Czajkowsky et al., 2004*). However, extensive equilibrium MD simulations (>0.3 μs) of a PFO monomer without the TMHs shows that the D2 domain remains in an extended conformation in the absence of the TMHs interactions, fluctuating in height by only a few Å (2.9 Å RMS height deviation) (*Figure 2A*). Thus, the loss of contact with the TMHs alone is insufficient to trigger the conversion of the D2 domain into a compact structure, a conclusion consistent with recent work showing that the TMHs in the thermally trapped PFO prepore are unfolded with significant conformational freedom (*Sato et al., 2013*) while cryo-electron microscopy (cryo-EM) images of a related cytotoxin, pneumolysin, showed that the D2 domain in this state is in its extended conformation (*Tilley et al., 2005*).

Closer inspection of this prepore conformation, in fact, reveals that fully extended TMHs are of a sufficient length to contact the bilayer surface (*Figure 2—figure supplement 1*). This extended conformation exposes hydrophobic residues in the TMHs to water, and so the membrane contact would generate a bilayer-directed hydrophobic force on the TMHs, driving them toward the membrane

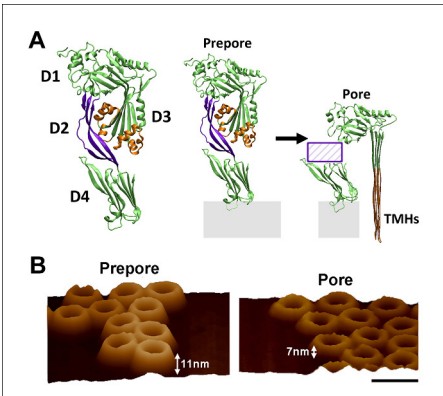

**Figure 1.** Structure of PFO and the known structural changes during pore formation. (**A**) The water-soluble monomer consists of four domains, D1 to D4. (**B**) AFM images showing the large height difference between prepore and pore complexes. Scale bar: 50 nm.

interior. Such a hydrophobic force exerted on a water-exposed transmembrane region has been reported in the folding of bacteriorhodopsin (*Kessler et al., 2006*) and implicated in the translocon-mediated membrane integration of transmembrane helices (*Ismail et al., 2012*).

To examine the effect of this bilayer-directed force on the conformation of the protein, we employed steered MD simulations, applying a downward constant force specifically on the TMHs. These calculations reveal that the most significant structural effect of this force is the collapse of the D2 domain (*Figure 2B*), with the rest of the structure remaining largely unchanged (*Figure 2—figure supplement 2*). That is, the tensile force on the TMHs is effectively transmitted through the protein to generate a compressive stress across the D2 domain that can catalyze its collapse. During the simulations, after modest initial changes in conformation, the protein structure remained relatively stable owing to the presence of an energy barrier that prevented further conformational changes. Finally, this barrier was overcome, and the height dropped rapidly to a value ~40 Å lower than its initial height (*Figure 2B*). In this final conformation, the lower portion of the D2 domain is still a β-sheet and lies almost perpendicular on the surface of the D4 domain, pivoting at Lys381. More importantly, the D2 domain remains in this compact conformation even after the force is relieved (for >40 ns) (*Figure 2—figure supplement 3*), suggesting that this conformation is a relatively stable state. The D2/D4 interface in this conformation covers 842 Å (*Kellermayer et al., 1997*) similar to that observed for interfacial contacts in oligomeric proteins (*Miller et al., 1987*), and the buried hydrophobic residues (Phe75, Tyr389, Tyr415) in the interface (*Figure 2C*) likely contributed to the stability of this compact state. As these residues are highly conserved in the cholesterol-dependent cytolysins (CDC) family (*Supplementary file 1*), it is possible that this structure is a common feature of other CDC toxins. For these simulations, the applied force was 250 pN; different forces consistently produced similar results, but the time scale was significantly longer at lower forces (*Figure 2—figure supplement 4*). Overall this collapsed structure is consistent with the electron-density profile of the pneumolysin pore by cryo-EM (*Tilley et al., 2005*) (*Figure 2—figure supplement 5*).

To further understand the energy associated with this transition, extensive adaptive biasing force (ABF) simulations (*Chipot and Hénin, 2005*) yielded the energy landscape along the reaction coordinate, the height of the D3 domain. This landscape (*Figure 2D*) exhibits a broad minimum at heights associated with the prepore conformation (from −10 to 8 Å) and a local minimum at heights associated with the pore conformation (centered at ~40 Å), separated by an energy barrier of ~23 kcal/mol (~38 $k_BT$) from the prepore minimum. Thus, the extended structure of the D2 domain is indeed associated with the energetically minimized conformation of PFO. Of note, there is a sharp rise in the energy at ~25 Å, where the energy increases by ~17 kcal/mol over a distance of ~6 Å. Such a barrier height would require a time frame of days to be overcome by thermal fluctuations alone (see 'Materials and methods'), far longer than the observed timescales of the prepore-to-pore transition (*Hotze et al., 2001*). Thus, this is the primary barrier that must be reduced by the force generated by the bilayer-TMHs interaction in order to drive PFO into its pore conformation.

As the aforementioned steered MD simulations revealed that it is, ultimately, a compressive stress across the D2 domain that is responsible for its collapse, we reasoned that this process could be experimentally probed with compressive forces applied by an AFM probe on individual complexes if the tensile forces owing to the bilayer-TMHs interaction could be prevented. In this regard, a mutant protein, PFO[G57C-S190C], has been identified that spontaneously assembles into prepore complexes on the membrane like the wild-type protein but does not proceed to the pore state owing to a disulfide bridge that inhibits the restructuring of the TMHs (*Figure 3A*) (*Czajkowsky et al., 2004*; *Hotze et al., 2001*). When this constraint is removed by adding Dithiothreitol (DTT), this mutant

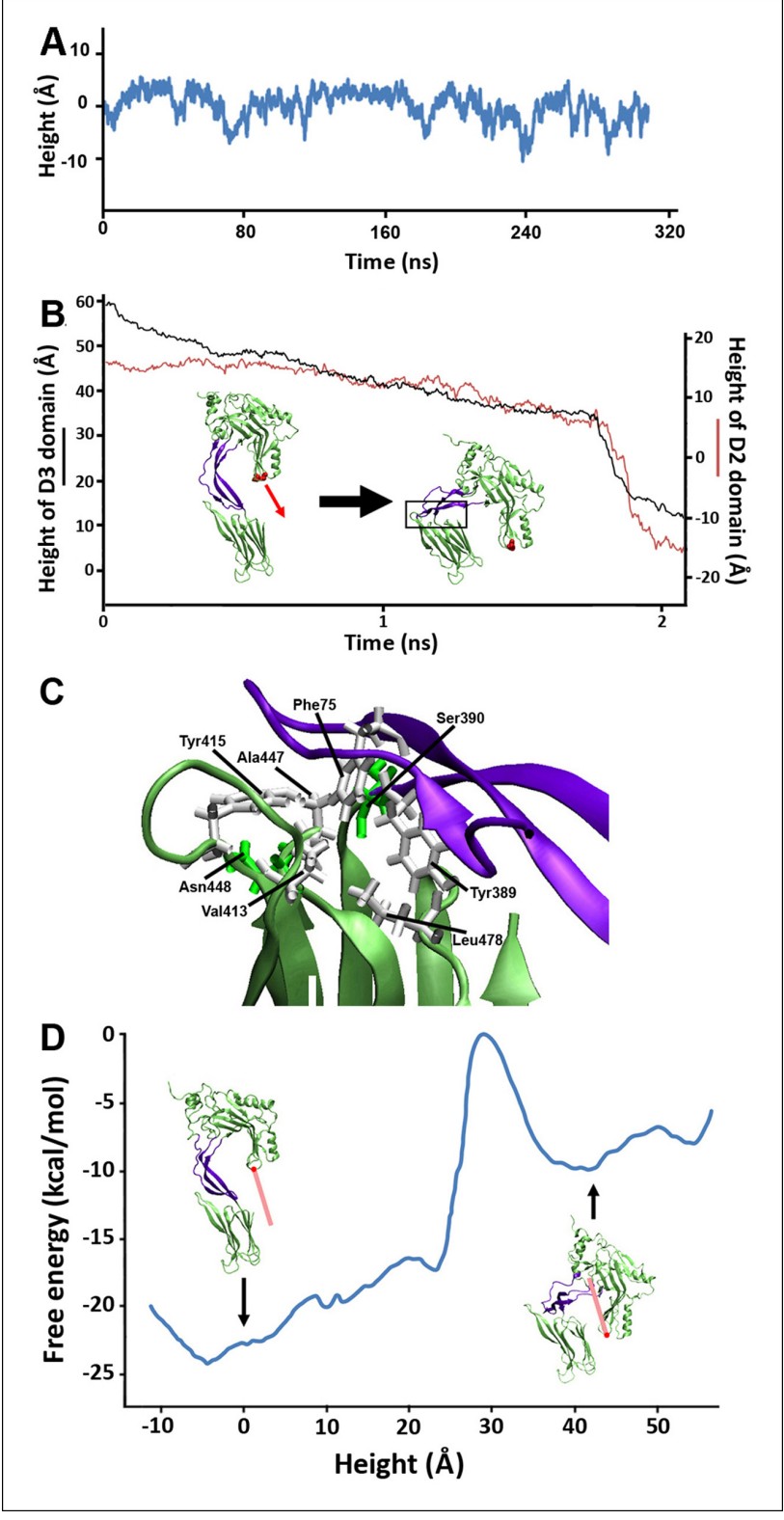

**Figure 2.** Steered MD simulations and energy landscape associated with the membrane-directed descent of the D3 domain. (**A**) The change in height of PFO during extended equilibrium simulations following the removal of the TMHs. (**B**) The D2 domain restructures as a result of a downward force (here 250 pN) applied to the D3 domain. Inset: the initial and final structures of the steered MD simulations. (**C**) Detailed view of the D2/D4 interface (the

*Figure 2. continued on next page*

*Figure 2. Continued*

boxed region in [B]). (D) Potential of mean force profile calculated from ABF simulations using the height of the D3 region (specifically the red sphere) as the reaction coordinate.

The following figure supplements are available for Figure 2:

**Figure supplement 1.** Fully extended TMHs are sufficiently long to touch the membrane surface from their position in the prepore.

**Figure supplement 2.** Structural changes in each domain during the steered MD simulations described in *Figure 2*.

**Figure supplement 3.** Height of the D2 domain following the release of the constant force on the D3 domain at the end of the steered MD simulations described in *Figure 2*.

**Figure supplement 4.** Steered MD simulations with different applied forces and force directions.

**Figure supplement 5.** Overlap between the electron density profile from cryo-EM images of the pneumolysin pore (*Tilley et al., 2005*) and the PFO structure at the final stages of the steered MD simulations.

quickly converts into a native-like pore (*Czajkowsky et al., 2004*; *Hotze et al., 2001*). Thus, this mutant enables investigation of just the energy barrier of the D2 collapse under compressive forces, without any complicating effects associated with the D3-bilayer interaction (*Figure 3B* and *Figure 3—figure supplement 1*). Steered MD simulations confirmed that under compressive forces mimicking those that would be applied with AFM, the D2 domain of this mutant recapitulates the same collapse as that of the D2 domain under tensile forces applied to the TMHs (*Figure 3—figure supplement 2*). Indeed, direct application of constant compressive forces with an AFM tip to individual PFO$^{G57C-S190C}$ prepore complexes reproducibly induced the characteristic height reduction in the absence of any reducing agent, leaving the general morphology of the complexes otherwise unchanged (*Figure 3C* and additional images are presented in *Figure 3—figure supplement 3*). We note that this pore-like structure does not revert back to the prepore conformation upon removing the applied force, indicating that this compact structure is relatively stable, consistent with the MD simulations. We further note that this applied force (110 pN/monomer) is >10-fold smaller than the force required to break a covalent bond over this time period (see 'Materials and methods') (*Grandbois et al., 1999*).

As expected for a thermally driven transition, this force-catalyzed conversion is dependent on the magnitude and the duration of the applied force (*Figure 3D*). Assuming a simple, single barrier model commonly employed in force spectroscopic experiments (*Hu and Li, 2014*; *Evans, 2001*; *Fernandez et al., 2010*), the probability to induce the pore-like state under an applied force, *F*, for a pulse period, *t,* is given by

$$P_{pore} = 1 - e^{-k_f t} \tag{1}$$

where

$$k_f = k_o \times e^{\frac{F x_\beta}{k_B T}} \, and \, k_o = A \times e^{\frac{-\Delta G_o}{k_B T}} \tag{2}$$

and $x_\beta$ is the reaction coordinate distance to the energy barrier peak from the minimum, $\Delta G_o$ is the barrier height, *A* is the attempt frequency, $k_B$ is Boltzmann's constant, and *T* is the temperature. Each 1s and 5s dataset is well described by the same single barrier model (*Figure 3—figure supplement 4*). Thus, we simultaneously fitted both datasets to the above equations, yielding $x_\beta$ = 5.1 ± 0.8 Å and $\Delta G_o$ = 16.3 ± 0.6 kcal/mol (see 'Materials and methods'). This energy barrier is smaller than that observed in previous single molecule protein domain unfolding experiments (*Fernandez et al., 2010*; *Dietz and Rief, 2008*), as might be expected since the D2 domain is not unfolded during this transition. Importantly, there is excellent agreement between the experimental values and the aforementioned ABF calculations (6 Å and 17 kcal/mol, respectively), thereby providing quantitative support for the model.

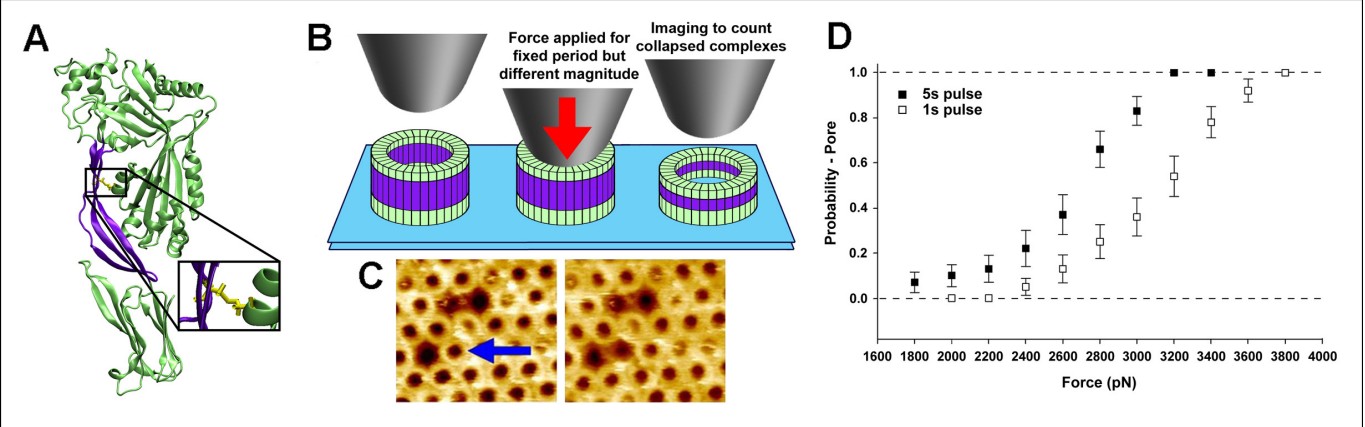

**Figure 3.** Single molecule compression provides a direct measure of the energy landscape of the D2 collapse. (**A**) PFO$^{G57C-S190C}$ prepore-trapped mutant. (**B**) Schematic of the experiment. (**C**) A force of 110 pN/monomer (~4000 pN/complex) applied for 1 s to a single prepore complex (blue arrow) catalyzed the collapse to the pore-like height. Image size: 200 x× 220 nm$^2$. (**D**) Force and time dependence of the AFM probe-induced D2 collapse.

The following figure supplements are available for Figure 3:

**Figure supplement 1.** Scanning electron micrograph of the AFM tip used in these experiments shows a symmetric apex with a radius of ~25 nm, consistent with manufacturer's specifications.

**Figure supplement 2.** Steered MD simulations reveal the collapse of the D2 domain under the application of compressive force to PFO$^{G57C-S190C}$.

**Figure supplement 3.** Additional AFM images showing the forced collapse of prepore-locked PFO$^{G57C-S190C}$ complexes.

**Figure supplement 4.** The AFM compressive data is well described by a two-state, single energy barrier model.

Since the hydrophobic force generated by the TMHs-bilayer interaction is critical for the prepore-to-pore transition, any weakening of this interaction is expected to cause a measurable reduction in the rate of transition. Therefore, we investigated the prepore-to-pore transition in the presence of a detergent (n-dodecyl-β-D-maltoside, DDM), since detergent binding to hydrophobic residues in the TMHs should reduce this hydrophobic force on the TMHs (*Figure 4A*). As shown in *Figure 4B*, incubating DDM with the mutant PFO$^{G57C-S190C}$ prepore complexes, followed by the addition of DTT, prevents conversion to the pore-like structure. This effect was dependent on the concentration of detergent, with a maximal inhibition observed with nearly 20 μM DDM (*Figure 4B*), a similar concentration as that observed for binding to exposed hydrophobic regions in other proteins (*Kragh-Hansen et al., 2001*).

In summary, the combination of single molecule measurements and MD computations has enabled direct identification of the essential role of intra-protein forces in driving the PFO complex into its final pore conformation (*Figure 4C*). We speculate that such intra-protein forces may also be a critical factor in driving coordinated conformational changes between distant domains in many other systems, which are presently not fully characterized mechanistically (*Cui and Karplus, 2008*; *Changeux and Edelstein, 2005*). These results also demonstrate that single molecule compressive force spectroscopy can be an effective means by which previously inaccessible physical mechanisms of conformational coordination within membrane proteins in particular may now be probed and quantified.

## Materials and methods

### Equilibrium MD simulations

The initial structure for the equilibrium simulations was the atomic model of PFO (pdb: 1PFO) (*Rossjohn et al., 1997*). All systems were solvated in TIP3 water in 0.15 M NaCl and minimized and equilibrated using VMD/NAMD and the CHARMM 27 force field (*Humphrey et al., 1996*;

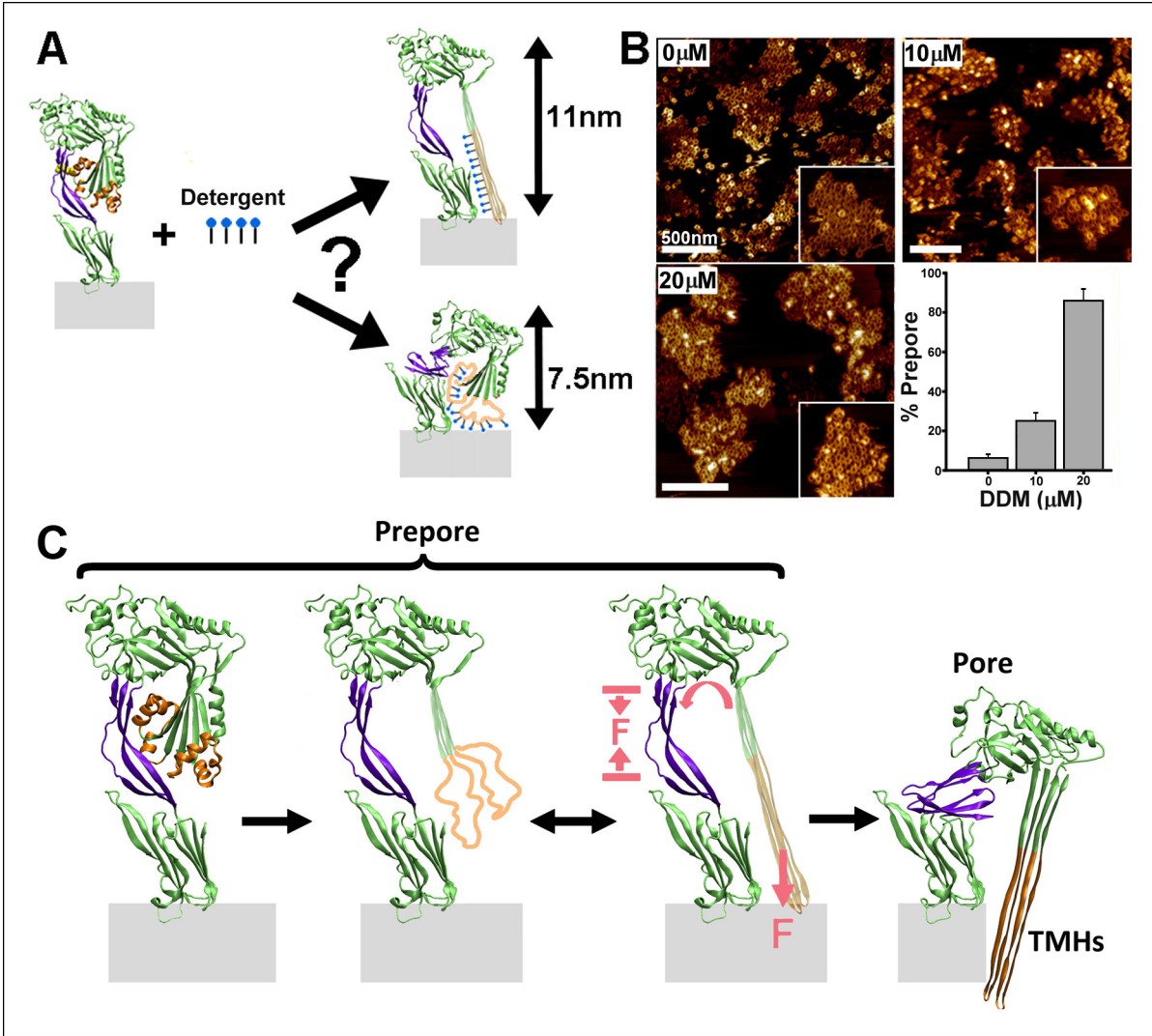

**Figure 4.** Effect of detergent on the prepore-to-pore transition and final model. (**A**) Potential outcomes of binding of detergent to hydrophobic residues within the TMHs. (**B**) AFM images showing the inhibition of the pore-like height with increasing concentration of the detergent, DDM. Inset image size: 550 x× 550 nm². (**C**) Schematic model of the mechanism by which conformational changes in the D2 and TMHs are coordinated. Compressive intra-protein stresses generated by the bilayer-directed forces on the unfolded TMHs catalyze the restructuring of the D2 domain to its collapsed conformation in the pore.

Mackerell, 1998; Phillips et al., 2005).

Langevin dynamics were employed to maintain a constant temperature of 310 K and a Nose-Hoover Langevin piston was used to maintain a constant pressure of 1 atm. The particle mesh Ewald algorithm was employed to treat electrostatic interactions, and the van der Waals interactions were treated with a cut-off of 12 Å. The integration step was set to 2 fs. For the simulations without the TMHs, the residues between Tyr187 to Val221 and Thr283 to Ser316 were removed and the remaining regions resealed. The height of the protein was determined from the distance between the centers-of-mass of the $C_\alpha$ backbone atoms for Lys127/Pro128 and Ala401/Tyr402.

### Steered MD simulations

The steered MD simulations were performed using the structure of PFO without the TMHs to avoid complications arising from unfolding of the TMHs and their contact with the other regions of the protein. In each simulation, the lower loop regions of domain 4 (residues 399 to 404, 434 to 438, 459 to 468, and 489 to 493) were held fixed to mimic their vertically immobile association with the

membrane surface during pore formation. A constant force was applied to the $C_\alpha$ atoms of residues Met222 and Ser317 following the standard NAMD protocol. The height of the D3 domain was determined from the positions of the $C_\alpha$ atoms of Met222 and Ser317. The height of the D2 domain was determined from the locations of the $C_\alpha$ atoms of Thr86 and Lys378. The direction of force application for most simulations was chosen as the shortest path from the $C_\alpha$ atoms of Met222 and Ser317 to the expected location of the membrane surface, avoiding direct collision with the D4 domain. In particular, this direction corresponds to a ~15° tilt from the vertical (the long axis of PFO), away from the protein, as depicted in *Figure 2—figure supplement 1A*. Other directions were also investigated, including the ~10° tilt direction depicted in *Figure 2—figure supplement 1B*. For the simulations with an applied compressive force, we studied the mutant protein, PFO$^{G57C-S190C}$. The downward force was applied to the top of the protein by a section of a carbon nanotube (generated with VMD), as shown in *Figure 3—figure supplement 2*.

## ABF MD simulations

The sampling space was limited to the separation of two centers of mass, one located in the D3 domain and the other near the bottom of the D4 domain, applying a restraining harmonic potential to the $C_\alpha$ atoms of Met222 and Ser317 to constrain the motion along the reaction coordinate such that it changed only vertically. The direction of this pathway is approximately that depicted in *Figure 2—figure supplement 1A*. One center of mass included the $C_\alpha$ atoms of Leu396 and Val484 and the other included the $C_\alpha$ atoms of residues 184, 185, 223, 224, 280, 281, 318, and 319. In addition, as with the steered MD simulations, the lower loop residues (residues 399 to 404, 434 to 438, 459 to 468, and 489 to 493) were held fixed during these calculations. Two different ABF runs were performed: first with a distance range between the aforementioned centers-of-mass of 1.5 Å and then piecing together these results as the initial potential of mean force profile for calculations with distance ranges of 5 Å. The latter were performed to guarantee accuracy of the calculated profile, particularly at the junction between neighboring 1.5 Å segments. The size of each ABF sampling bin for all calculations was 0.1 Å. All simulations, run within NAMD, were run long enough to observe the convergence of the system to ensure accurate sampling of the free energy profile: at the end of the second ABF run, every ABF 0.1 Å bin had been sampled over 100,000 times.

## Materials

The mutant PFO$^{G57C-S190C}$ was produced and purified as before (*Hotze et al., 2001*). Egg phosphatidylcholine (eggPC), cholesterol (chol), and 1,2-bis(10,12-tricosadiynoyl)-*sn*-glycero-3-phosphocholine (diynePC) were purchased from Avanti Polar Lipids (Alabaster, AL, US). n-dodecyl-β-D-maltoside (DDM) was purchased from Anatrace (Maumee, OH, US). All other chemicals were purchased from Sigma (St. Louis, MO, US).

## AFM sample preparation and imaging

The supported membranes containing mutant PFO$^{G57C-S190C}$ were formed by sequentially depositing two separately prepared lipid monolayers onto a mica substrate, followed by the injection of the protein into small Teflon wells, as previously detailed (*Czajkowsky et al., 2004*). For the single molecule compressive force experiments, the composition of the first monolayer (facing mica) was eggPC, and the second monolayer was eggPC:chol at 50:50 mol%. For the experiments with the detergent, supported bilayers of a similar composition were dissolved too quickly following the addition of detergent to enable investigation. Instead, for these experiments, the first monolayer was composed of diynePC and the second monolayer was eggPC:chol (50:50 mol%). Following deposition of the diynePC monolayer onto mica, the sample was irradiated with UV light (Bio-Rad) for 10 min to induce cross-linking of the lipids. As is evident from *Figure 4B*, the PFO$^{G57C-S190C}$ complexes proceed to their pore-like height in the presence of DTT, just as in bilayers where the first monolayer is composed of eggPC (*Czajkowsky et al., 2004*). For all experiments, the final concentration of the protein in the well was ~15 µg/ml, and the buffer in the well consisted of buffer A (10 mM sodium phosphate, pH 7). After incubating for 45 min, the sample was extensively washed and then, for the compressive force experiments, imaged in the AFM under buffer A. For the experiments with detergent, the sample was first washed in buffer B (10 mM HEPES, 0.1 M NaCl, 25 mM CaCl$_2$, pH 7.5), then washed and incubated for 5 min in buffer B with DDM (at the concentrations indicated in

*Figure 4B*), then washed and incubated for 10 min in buffer B with DDM and 2.5 mM DTT, then washed and incubated for 10 min in buffer B with DDM and 2% glutaraldehyde, followed finally by an extensive wash in buffer B, a final incubation in 20 mM glycine, pH 6, and a final wash and imaging in buffer B. The chemical fixation was necessary because stable imaging of the protein complexes directly in the detergent solution was not possible. The histogram data presented in *Figure 4B* was determined from imaging five different regions in each of three different samples at each condition. Imaging was performed in the contact mode with a Nanoscope II AFM (Bruker - Digital Instruments, Santa Barbara, CA, US) using oxide-sharpened 'twin-tip' $Si_3N_4$ cantilevers with a spring constant of 0.06 N/m. The spring constant was verified by measuring the thermal induced oscillation. The typical scan rate was 9 Hz, and the imaging force was minimized to 0.1 nN. The piezoscanner (14 mm, D scanner, Bruker - Digital Instruments, Santa Barbara, CA, US) was calibrated using a variety of samples including mica and the cholera toxin B subunit. Scanning electron microscopy of the AFM probe (shown in *Figure 3—figure supplement 1*) was obtained with a JEOL 7800F microscope (Peabody, MA, US).

## Single molecule compression with AFM

Compressive forces have been previously applied with AFM to probe mesoscopic mechanical properties of various biomaterials (*Lee et al., 2013*; *Kodama et al., 2005*). In our experiments, after obtaining an image of the sample at small applied forces in scan sizes of 300 to –400 nm, the tip was positioned within the center of a given complex by zooming in with progressively smaller scan sizes (generally 2), arriving at a sample location with the complex of interest in the center of a scan size of ~80 nm. Once one-half of the complex was imaged, we immediately zoomed in on its center, set the scan size to zero, and then applied the larger constant force for a specific duration (1 or 5 s). The force was then immediately reduced to detach the tip from any contact from the sample, the scan size was adjusted to ~500 nm, and then the tip was re-engaged at smaller forces to obtain an image of the sample to determine whether or not the larger applied force catalyzed the height reduction in the complex. In this way, the applied force was equally distributed on each of the monomers within the complex. No complex was probed more than once. We measured the time between the final zoom-in step and the application of force to be less than 5 s. We measured the extent of lateral drift in our system to be less than 0.03 nm/s, which is similar to published reports (*King et al., 2009*). Thus, we expect to not have drifted by more than 0.15 nm from the center of the complex prior to force application. As the complex is 35 nm in diameter, this drift imparts only a slight deviation to an equal force on all subunits. Although wild-type PFO complexes and other CDC toxins form a range of sizes, from incomplete arcs to complete rings (*Czajkowsky et al., 2004*; *Sonnen et al., 2014*), these prepore mutant PFO$^{G57C-S190C}$ complexes formed almost exclusively complete rings of an identical size. Only the complete rings were studied here. Based on the radial periodicity of the monomers in the complete rings (*Czajkowsky et al., 2004*) and high-resolution images of the rings (data not published), the complete rings contain 36 subunits. Thus, in the text, we referred to the applied force either with respect to the entire complex or to each monomer (dividing the total applied force by 36). As such, the total applied force range used here (1800 pN to –3800 pN) corresponds to 50 pN to –106 pN per monomer, which is at least an order of magnitude smaller than the force needed to break a covalent bond over this time frame (1400 pN to –2000 pN) (*Grandbois et al., 1999*). On average, more than 30 individual measurements were performed at each applied force, resulting in more than 600 individual single molecule measurements obtained from tens of samples and using tens of different AFM cantilevers.

## Data analysis

To obtain a measure of the energy barrier height ($\Delta G_o$) and the reaction coordinate distance ($x_\beta$) from the AFM data, we first sought evidence that each of the two datasets (the 1s and 5s data) was individually consistent with a similar single barrier model. To this end, *Equation 1* was linearized with respect to force as

$$y(F) = \ln\left(\ln\left(\frac{1}{1 - P_{pore}}\right)\right) = \ln(k_0 t) + \frac{F x_\beta}{k_B T} \qquad (3)$$

and then we determined whether the fits of each of the 1 s and 5 s datasets (excluding the data for which $P_{pore}$ is equal to 1 to avoid division by zero) to this equation yielded a similar value for the slope (or $x_\beta$), using the force, $F$, per monomer. As shown in *Figure 3—figure supplement 4*, each of the datasets was indeed described by this equation with similar values of $x_\beta$: $x_\beta = 4.6 \pm 1.9$ Å for the 1 s dataset and $x_\beta = 4.4 \pm 1.8$ Å for the 5s dataset. With this assurance, we then globally fit the entire 1 s and 5 s datasets to *Equation 1*, yielding $x_\beta = 5.1 \pm 0.8$ Å and $k_o = 1.7 \times 10^{-5} \pm 2.8 \times 10^{-5}$/s. Using an attempt frequency of $10^7$/s, consistent with previous force spectroscopy studies (*Yu et al., 2012*), we obtain (using *Equation 2*): $\Delta G_o = 16.3 \pm 0.6$ kcal/mol.

For the barrier height obtained from the ABF calculations (17 kcal/mol), using a similar attempt frequency and *Equation 2* yields $k_o = 6.9 \times 10^{-6}$/s, which corresponds to a (force-free) lifetime of ~1.7 days.

## Acknowledgements

We thank Professor Rod Tweten for generously providing the mutant PFO used in these experiments. We thank Dr. Jingfang Wang for assistance with preliminary computations. This work was supported by the NSFC (91129000, 11374207, 31370750, 21273148, 91229108, and 21303104), the Science and Technology Commission of Shanghai Municipality (15142201200), Shanghai Jiao Tong University (2012MS58), and the KC Wong Education Foundation (HK).

## Additional information

### Funding

| Funder | Grant reference number | Author |
|---|---|---|
| National Natural Science Foundation of China | 91129000 | Daniel M Czajkowsky<br>Jielin Sun<br>Yi Shen<br>Zhifeng Shao |
| KC Wong Education Foundation | | Zhifeng Shao |
| National Natural Science Foundation of China | 11374207 | Daniel M Czajkowsky<br>Jielin Sun<br>Yi Shen<br>Zhifeng Shao |
| National Natural Science Foundation of China | 31370750 | Daniel M Czajkowsky<br>Jielin Sun<br>Yi Shen<br>Zhifeng Shao |
| National Natural Science Foundation of China | 21273148 | Daniel M Czajkowsky<br>Jielin Sun<br>Yi Shen<br>Zhifeng Shao |
| National Natural Science Foundation of China | 21303104 | Daniel M Czajkowsky<br>Jielin Sun<br>Yi Shen<br>Zhifeng Shao |
| National Natural Science Foundation of China | 91229108 | Daniel M Czajkowsky<br>Jielin Sun<br>Yi Shen<br>Zhifeng Shao |
| Science and Technology Commission of Shanghai Municipality | 15142201200 | Daniel M Czajkowsky<br>Jielin Sun<br>Yi Shen<br>Zhifeng Shao |
| Shanghai Jiao Tong University | YG2012MS58 | Daniel M Czajkowsky<br>Jielin Sun<br>Yi Shen<br>Zhifeng Shao |

The funders had no role in study design, data collection and interpretation, or the decision to submit the work for publication.

## Author contributions

DMC, Conceived and designed the experiments, performed most of the experiments, and wrote the manuscript; JS, YS, Contributed to some of the experiments; ZS, Conceived and designed the experiments, supervised all of the research, and wrote the manuscript

## Additional files

### Supplementary files

• Supplementary file 1. Sequence similarity of the hydrophobic residues in PFO at the D2/D4 interface in the collapsed D2 conformation among members of the CDC family. The proteins were aligned using BLAST. Shown in parentheses are the residue numbers for each protein.

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
