## [Decision Letter]

Thank you for submitting your work entitled "Single molecule compression reveals intra-protein forces drive cytotoxin pore formation" for peer review at *eLife*. Your submission has been favorably evaluated by Michael Marletta (Senior editor) and three reviewers, one of whom is a member of our Board of Reviewing Editors.

The reviewers have discussed the reviews with one another and the Reviewing editor has drafted this decision to help you prepare a revised submission.

Czajkowsky et al. present a study that combines MD simulations and single-molecule AFM experimental data to analyze Perfringolysin O (PFO) pore-forming protein. The conformational changes responsible for the pre-pore to pore transition were previously known. The authors themselves published an article in 2004 in EMBO journal that uses the same di-sulfide-linked mutant in combination with AFM imaging with or without DTT to observe the shrinking of the pore heights due to pore formation. Here, the new contribution is that compressive AFM is used to induce pore formation. Also, steered MD simulations are performed to complement experiments. Importantly, they are able to obtain the transition probability as a function of force magnitude and duration, allowing them to fit the data to a two state model and deduce the distance to the transition state (and less clearly) the free energy barrier. If indeed this is the first time such analysis has been performed, this would be a significant contribution regardless of the novelty and biological significance of the findings themselves. In the physiological context, the force will be applied between D3 and the membrane, largely driven by hydrophobic force, and the compressive force is proposed to mimic this force, which is reasonable. MD simulations provide molecular accounting and some quantitative comparisons. The experiments were performed in an expert fashion, and the data is highly original and of high quality. Because we are not aware of any prior publication measuring a certain rate as a function of compressive force from a single molecule, this work could be a significant contribution even without important biological insights as long as the authors can address important technical issues raised, including a major concern about the validity of the construct used for the study. In addition, the work presents possible mechanisms underlying the transition from prepare to pore of PFO.

Essential revisions:

1) The pore-forming complex used in the AFM studies is a covalently-locked disulfide mutant that known from the authors' previous work not to form pores. They also state the compressive forces applied by the AFM tip are too low to break these disulfide bonds. Based on these two facts, it follows that the mechanism of pore collapse in their AFM experiment must be very different than the natural conformational change in the protein. This undermines their claim that "targeted application of compressive forces can recapitulate a critical step" of the pore-formation. How can they obtain a pore-like conformation in the presence of disulfide bond? There is no evidence presented that shows that a pore-like conformation can be obtained without breaking the disulfide bond between D2 and D3. We suggest that the authors perform MD simulations with the disulfide bond present and present evidence that the force results in pore-like conformations of D2 and D3 even in the presence of disulfide bond.

2) The authors assume that the compressive force is applied equally to all subunits. But they do not provide any evidence. What if the conformational change occurs efficiently only when the force is applied to primarily one subunit? Is their AFM free of lateral drift enough to be sure that the tip is pushing down in the middle of the pore? What is the AFP tip is not symmetrically shaped? What was the tip radius of the cantilever they used for AFM imaging and compression experiments? If the authors could please provide an SEM image of the cantilever tip so the reader can make a comparison to the pore size, it would improve the manuscript.

3) The tensile pulling geometry in the MD simulations does not mimic that found in the experiment, where compression is applied to the pore from above. The specific geometry will of course play a major role in determining the mechanical response of the protein pores. Additionally, the lipid membrane is not simulated in the MD trajectories. These dissimilarities between MD and experiment are too large to make any quantitative comparisons between the two results.

4) Perhaps the biggest issue with the MD simulations is that the authors performed MD simulations using one monomer, not as part of the complex. How does the conformational change in one PFO trigger the pre-pore to pore transition for all the PFOs in the pore complex? This is a major limitation because subunit interactions are likely to be important in determining the force response and conformational changes.

---

## [Author Response]

*1) The pore-forming complex used in the AFM studies is a covalently-locked disulfide mutant that known from the authors' previous work not to form pores. They also state the compressive forces applied by the AFM tip are too low to break these disulfide bonds. Based on these two facts, it follows that the mechanism of pore collapse in their AFM experiment must be very different than the natural conformational change in the protein. This undermines their claim that "targeted application of compressive forces can recapitulate a critical step" of the pore-formation. How can they obtain a pore-like conformation in the presence of disulfide bond? There is no evidence presented that shows that a pore-like conformation can be obtained without breaking the disulfide bond between D2 and D3. We suggest that the authors perform MD simulations with the disulfide bond present and present evidence that the force results in pore-like conformations of D2 and D3 even in the presence of disulfide bond.*

There may be some misunderstanding of a point that we should have made more explicit. Based on numerous lines of evidence (Cell 97, 647 (1999), EMBO J 23, 3206 (2004), Cell 121, 247 (2005), Nat Chem Biol 9, 383 (2013)), it has been well established that the significant compaction of the D2 domain is a critical conformational change of pore formation. The unfolding of the TMHs in the D3 domain are not directly coupled structurally to that of the collapse of the D2 domain, thus the mechanism of D2 conversion remained unresolved prior to our work. That is, the objective of our study was to determine how the D2 domain is driven into its compact conformation, not the insertion of the TMHs into the lipid bilayer per se. As the reviewer correctly pointed out, with the disulfide bridge intact, these mutants cannot form transmembrane pores. But this mutant protein enabled testing of the predictions from the MD simulations of the compressive force-catalyzed collapse of the D2 domain without any complicating effects associated with the D3-bilayer interaction. We have now included a figure (Figure 4) to further clarify this point. Additional text of explanation is also included in Results and Discussion, third and fifth paragraphs,.

As suggested, we have now performed MD simulations of a PFO protein with the disulfide bond present under a compressive force mimicking that applied during the AFM experiments. Now included as (new) Figure 3—figure supplement 2, these simulations clearly show that, under compressive forces mimicking those that would be applied with AFM, the D2 domain of this mutant recapitulates the same collapse as that of the D2 domain under tensile forces applied to the TMHs (see also Figure 5). Also, the protein maintains this compact structure even after release of the compressive force, similar to the previous simulations (Figure 3—figure supplement 2). We have rephrased our previous text and included additional discussion describing these results in Results and Discussion, fifth paragraph, and in the subsection “Steered MD simulations”. We are grateful for this suggestion, as these results further strengthened our main conclusions.

Author response image 1.Comparison of the structures of the D2 domain and the D2/D4 interface, following the application of compressive or tensile forces.Shown are snapshots of the structures 40 ns after the release of the applied force.**DOI:**
http://dx.doi.org/10.7554/eLife.08421.017

*2) The authors assume that the compressive force is applied equally to all subunits. But they do not provide any evidence. What if the conformational change occurs efficiently only when the force is applied to primarily one subunit? Is their AFM free of lateral drift enough to be sure that the tip is pushing down in the middle of the pore? What is the AFP tip is not symmetrically shaped? What was the tip radius of the cantilever they used for AFM imaging and compression experiments? If the authors could please provide an SEM image of the cantilever tip so the reader can make a comparison to the pore size, it would improve the manuscript.*

We appreciate these comments. Indeed, we cannot entirely exclude the possibility of a slightly unequal distribution of applied forces in some of the measurements. However, the data presented were obtained from tens of samples probed with many batches of tips. As such, the ensemble of such measurements should average out such “outliers”. In fact, overall, the data are well-behaved, suggesting that unequal applications of force did not constitute a significant portion of our data.

Further, we should have been more explicit about our experimental protocol regarding the effect of lateral drift during the period of force application. With this protocol, the time between the final zoom-in step and the application of force was measured to be less than 5 s. We measured the extent of lateral drift in our system to be less than 0.03 nm per second, which is similar to published reports (Nanolett 9, 1451 (2009)). Thus, at most, we expect a drift of less than 0.3 nm from the center of the complex by the end of the force application. As the complex is ~35 nm in diameter, this drift, clearly, imparts only a slight deviation to an equal force on all subunits. We have now added this information to the Methods section, in the subsection “Single molecule compression with AFM”.

Finally, as suggested, we have obtained an SEM image of our AFM probes and include this as (new) Figure 3—figure supplement 1. The ultimate probe shape is essentially spherical at the apex and approximately 25 nm in radius, consistent with the manufacturer’s specifications. As this is roughly the same as the diameter of the pore, it is reasonable to assume that the applied force should be more or less equally applied to all subunits within the time frame of each data acquisition. We have now included this information to this figure legend and additional text to the Methods section: “Scanning electron microscopy of the AFM probe (shown in Figure 3—figure supplement 1) was obtained with a JEOL 7800F microscope (Peabody, MA).”

*3) The tensile pulling geometry in the MD simulations does not mimic that found in the experiment, where compression is applied to the pore from above. The specific geometry will of course play a major role in determining the mechanical response of the protein pores. Additionally, the lipid membrane is not simulated in the MD trajectories. These dissimilarities between MD and experiment are too large to make any quantitative comparisons between the two results.*

As described above, we have now performed MD simulations with compressive forces mimicking those in the AFM experiments, and these show a similar compaction of only the D2 domain, just as with the tensile forces. The reason for performing the tensile force MD simulations was the speculation that the tensile force on the TMHs in the actual pore-forming process generates a compressive force across the D2 domain. As such, the D2 would collapse under a compressive force regardless whether it is generated by the membrane insertion of the TMHs or by the AFM probe pushing the top of the protein. This is indeed the case as shown by the MD simulations. Also, we note that, based on AFM and cryo-EM data (EMBO J 23, 3206 (2004), Cell 121, 247 (2005)), the contact of the protein with the lipid bilayer does not change the structure of the D2 domain from that in the monomer. We thank the reviewers for this suggestion and the newly added MD results further supported our approach.

*4) Perhaps the biggest issue with the MD simulations is that the authors performed MD simulations using one monomer, not as part of the complex. How does the conformational change in one PFO trigger the pre-pore to pore transition for all the PFOs in the pore complex? This is a major limitation because subunit interactions are likely to be important in determining the force response and conformational changes.*

We agree with the reviewers that, in principle, it would be optimal to simulate the entire process, from the monomer binding to cholesterol, to oligomerization, to the release of the D3, to the compaction of the D2 domain, and finally to pore formation. But such simulations are clearly beyond our current computational capability. Furthermore, the interface between the subunits is known to involve only the D1 and D3 domains (and not the D2 domain), but the precise details of this interaction are largely not understood, and remain one of the main enigmas in the complete understanding of pore formation for this class of toxins. These limitations notwithstanding, it is well established that the compaction of the D2 domain is a critical step of the pore formation process, which is the central question we wish to answer in this paper: namely, through what mechanism is the D2 domain compacted? Previous work suggested that oligomerization is responsible for the separation of the D3 domain from contact with the D2 domain, leading to the unraveling of the TMHs. Their contact with the lipid bilayer is the driving force leading to the irreversible compaction of the D2 domain. The latter point is provided by the studies presented in this paper, and we are certainly keen to pursue answers to the other questions in future studies.